# Longitudinal Training and Workload Assessment in Young Friesian Stallions in Relation to Fitness, Part 2—An Adapted Training Program

**DOI:** 10.3390/ani13040658

**Published:** 2023-02-14

**Authors:** Esther Siegers, Jan van den Broek, Marianne Sloet van Oldruitenborgh-Oosterbaan, Carolien Munsters

**Affiliations:** 1Department of Clinical Sciences, Faculty of Veterinary Medicine, Utrecht University, Yalelaan 114, 3584 CM Utrecht, The Netherlands; 2Equine Integration, Groenstraat 2C, 5528 NS Hoogeloon, The Netherlands

**Keywords:** training, workload, performance, standardized exercise test, Friesian horses

## Abstract

**Simple Summary:**

Young Friesian stallions have to complete a 10-week training program (70-day test) for acceptance as a studbook-approved breeding stallion. Part one of this study showed that the original training program was too intense and led to reduced fitness. In part two of the study, the stallions performed fewer training sessions and fewer minutes of cantering per week. The effect of this adapted training program on the fitness of young Friesian stallions was evaluated. Sixteen stallions participated in part two of this study. Data were collected during the 6-week preparation period before the test as well as during the 70-day test. The total duration and time spent at each gait were measured during all training sessions. The horses performed three standardized exercise tests (SETs) in week 1 (SET-I), 6 (SET-II), and 10 (SET-III), where heart rate (HR) and plasma lactate concentration (LA) were measured. On average, the horses showed an increase in fitness at SET-III compared with SET-I, as indicated by decreased HR and LA levels during the same exercise. This study shows that young Friesian horses were at risk to being overtrained during the original training program, but the adapted training program resulted in an increase in the fitness level. A carefully chosen training program is necessary for improving the performance and welfare of (young) horses.

**Abstract:**

Young Friesian stallions have to complete a 10-week training program (70-day test) for acceptance as a breeding stallion. Part one of this study showed that the 70-day test was too intense and led to reduced fitness. In the present (part two) study, the effects of an adapted training program were studied. Training frequency and minutes of cantering per week were lowered compared with part one. The external workload (EW) of sixteen stallions (3.4 ± 0.8 years) was monitored during the 6-weeks before testing (preparation period) as well as during the 70-day test. Standardized exercise tests (SETs) were performed in week 1 (SET-I), 6 (SET-II), and 10 (SET-III) of the 70-day test, measuring heart rate (HR) and plasma lactate concentration (LA). Linear mixed effect models were used to analyze the EW, and the HR and LA related to the SETs. The EW increased from the preparation period to the 70-day test; this increase in the EW was lower compared with the original training program. The horses showed lower HR and LA levels in SET-III compared with SET-I. The young Friesian horses were at risk to be overtrained in part one, but the adapted training program resulted in increased fitness. This study shows that a carefully chosen program is necessary to improve the performance and welfare of (young) horses.

## 1. Introduction

Nowadays, the welfare of sport horses is under increasing scrutiny and criticism from the general public, but also from inside of the equine industry [1]. The Equine Ethics and Wellbeing Committee of the Fédération Équestre Internationale (FEI) performed several international surveys on people with and without an equine background. Seventy-five percent of the equestrian respondents were concerned about the use of horses in sport [1]; listed in the top ten of concerns were: ‘the other 23-hours’, ‘physical stress and injuries’, ‘overworking’, ‘pushed past limits’, and ‘age (too young)’.

An appropriate design of a training program contributes to the welfare of equine athletes [2]. The goal of training horses is to enhance performance, prepare them for the desired task, and prevent injuries from occurring [3]. Every given physical effort, and thus, every intense training session for horses and humans, leads to a disruption of the homeostasis in the cells and organs; this may lead to mechanical output and fatigue. In the recovery phase, damage is repaired and homeostasis is restored, and as a result, supercompensation and adaptation occur. The degree of fatigue and cellular damage depends on the magnitude of the training impulse. In the ideal situation, the next intense exercise session is performed after the adaptation and recovery phase. Repeatedly performing intense exercise before the recovery phase is finished increases the risk of overtraining [4,5,6,7]. Functional overreaching is defined as a short-term (up to two weeks) decrease in performance followed by supercompensation. Nonfunctional overreaching occurs when the performance decrease lasts for 3–4 weeks and is not followed by supercompensation. The term overtraining syndrome (OTS) is applicable to individuals undergoing an accumulation of training impulses with no recovery periods or ones that are too short, which results in a long-term (more than 3–4 weeks) decrement in performance capacity without relevant physiological and psychological symptoms [5,6,7]. In overtraining, there is an imbalance between training and recovery, often accompanied by non-training factors inducing mental or physical stress [4,5,6,7,8]. OTS may occur after several weeks of intense training; however, the etiopathogenesis of OTS in horses is relatively poorly understood [5]. In addition to reduced performance, equine and human studies have shown an increased risk of injuries in athletes experiencing inappropriate training loads [9,10].

In the equestrian industry, there is a lot of debate discussing the optimal age for horses to start training and what the optimal training approach should be. There are strong concerns from some horse owners, riders, and trainers that horses are starting their training when they are too young [1,11,12]. At the moment, there is a lack of scientific evidence on optimal training programs for young horses. However, more and more information is emerging around the load capacity of tissues and their dynamic response throughout a horse’s life. Studies in racehorses showed that thoroughbreds and standardbreds entering race training at the age of 2 years had more race starts, greater earnings, higher average racing distance, and longer racing careers than horses entering training later in life [13,14]. Training horses while they are young and growing seems to be beneficial for the production of a strong musculoskeletal system, which reduces the risk of injuries [13,14,15]. Scientific evidence on performance careers, injury risks, and longevity in relation to the initiating age of training in warmblood sport horses is currently not available.

At the moment, there is limited evidence on what the appropriate training program for a young warmblood sport horse is. Descriptive studies of training programs in different equestrian disciplines are available; however, most of these data are not related to performance [16,17,18]. Some studies of racehorses have analyzed the effect of a training program on (race) performance or injury occurrence [9,19,20,21,22,23,24]. Training frequency and alternating between high- and low-intensity training sessions have been shown to be an important factor in optimizing the training programs. In the study of de Graaf-Roelfsema et al., overtraining was induced in standardbreds by intense training (at ~80% of maximal HR) for 3–4 days a week and endurance training (at ~60% of maximal HR) three times a week for 6 weeks, compared with no overtraining when horses had four training sessions a week (two high-intensity and two endurance training sets) alternated with recovery days [25]. In the study of Lindner et al., racehorses had to perform two 5-min bouts at V_10_ (the horses’ speed at a LA of 10 mmol/L) with 5 min of walking in between every other day for 6 weeks. This was considered as intensive training. Horses showed signs of OTS, where their V_4_ and V_10_ reduced and the horses became uncooperative. When training with similar intensities but with two recovery days in between, the program led to improved fitness parameters [20,21]. Another study in young thoroughbreds showed increased fitness performance in exercise tests with intense training three times a week. These horses performed, after a warm-up, 2 min of cantering at the speed of 95% of their maximal oxygen consumption in normal oxygenated and in hypoxic conditions. The running time increased in these horses, showing improved fitness [26]. All of these findings are in line with the general training principle of supercompensation, where a training impulse should be followed by enough time for adequate recovery of that tissue [4,5,6,7]. Studies evaluating the effect of training programs in warmblood sport horses are currently sparce [10].

In the selection process of the licensing of breeding stallions for a studbook, young horses are evaluated on breed-specific traits, health, and performance by the specific studbook. Several studbooks in the Netherlands train their potential studbook stallions for several weeks at a studbook facility to evaluate their riding and/or driving performance, trainability, and other characteristics. Young Friesian stallions, selected by the Royal Friesian Horse Studbook (KFPS) are brought together annually at a training facility to be trained and educated for 70 days (70-day test). In part one of the study [27], the young stallions were trained according to this original studbook training program. The horses demonstrated reduced performance in standardized exercise tests (SETs) after 6 and 10 weeks of training, indicative of nonfunctional overreaching or overtraining. External workload, measured as the training duration per week, increased by 18% in the 70-day test compared with their normal training routine at home in the period before the 70-day test. Horses were trained 4.7 times per week, and training sessions were on average 144 min per week during the 70-day test. In the first and second SET, 7% and 63% of the participating horses exceeded HR_LA4,_ which is the HR where plasma lactate concentration is ≥4 mmol/L, in canter 2. In the third SET, the mean HR in canter 2 was 175 bpm and exceeding HR_LA4_ in all participating horses. This indicates that the canter was a high-intensity exercise for these young Friesian stallions.

The aim of this study was to evaluate the physiological effect of an adapted training program in young Friesian stallions participating in the 70-day test. In the adapted training program, the training frequency and duration of training sessions were reduced. High-intensity training was followed by one or two days of relative rest or low-intensity training. The time spent in canter was reduced, and cantering was only performed during the high-intensity training sessions.

## 2. Materials and Methods

### 2.1. Study Design

The study consisted of two parts: (I) the preparation period before the 70-day test and (II) the actual 70-day test (see Figure 1). The 70-day test is the final exam for Friesian stallions to become an approved KFPS studbook stallion at the KFPS. In the months before the 70-day test (preparation period), a large group of stallions (*n* = 43) was evaluated by official studbook judges during three specific selection days. Based on their total performance, the judges decided which stallions were allowed to enter the 70-day test. The studbook judges based their decision on selection criteria such as exterior traits, gaits, and character as well as genetic information and health assessments such as sperm quality, left laryngeal hemiplegia, and osteochondrosis dissecans [28]. In the end, 16 stallions were selected by the judges to participate at the 70-day test as their last test to become an approved studbook stallion. The inclusion criterion for the horses to participate in the study was being selected by the official studbook judges of the Friesian studbook (KFPS) to participate in the 70-day test.

#### 2.1.1. Preparation Period

The preparation period consisted of a 6-week period before the start of the 70-day test. During the preparation period, horses were trained by their individual rider and trainer in their normal environment. One of the selection days was held during the preparation period.

#### 2.1.2. 70-Day Test

During this part of the study (part two), the training program of the 70-day test was adapted in response to the results of part one of the study, where horses were trained according to the original studbook training program. In the adapted training program, the training frequency and duration of the training sessions were reduced. High-intensity training was followed by one or two days of rest (paddock turnout) or low-intensity training. During high-intensity training sessions under the saddle, horses performed canter exercise, but total time spent in canter was reduced to a maximum of 4 min per training session divided into several bouts of 1–2 min of cantering alternated with walking (referred to as ‘dressage training’). A high-intensity training session for driving exercise contained walking and trotting. The maximum duration of training sessions was 30 min. The standardized exercise tests (SETs) were also considered as high-intensity training. Low-intensity training sessions were lunging without cantering or riding without cantering. On days without training, horses were allowed in the paddock (see Table 1).

During the 70-day test, the horses’ characters, dressage and driving abilities, and health were evaluated. The horses were trained by 4 experienced riders and 2 experienced drivers. The horses were trained by the same rider as much as possible. Each stallion’s performance, behavior, stable manners, and occurrence of any veterinary problems were documented. Studbook judges assessed the horses twice for their studbook capacity in week 6 and 10; stallions could be eliminated from the 70-day test based on the judges’ score or as a result of an injury or other illness as assessed by a veterinarian. During the 70-day test, three standardized exercise tests (SETs) were performed in week 1, week 6, and week 10.

#### 2.1.3. Horses

Data were collected from privately owned young Friesian stallions. Sixteen stallions at the age of 3 years (*n* = 11), 4 (*n* = 3), or 5 (*n* = 2) years were selected by the KFPS to participate in the 70-day test. During the entire study (preparation period and 70-day test), horses were kept at individual stables and were fed an individual diet consisting of roughage and pellets. Water was provided ad libitum. The Animal Ethics Committee of Utrecht University approved all procedures (reference number 5204-1-04, approval date 24 June 2020). Written owner’s informed consent was also obtained for each horse.

### 2.2. Equipment

During the SETs in the 70-day test, HR recording was performed using an HR sensor (Polar V800^®^ Polar Electro Oy, Kempele, Finland). During the 70-day test, the duration of the exercise and the time spent in each gait were measured using a 9-axis motion sensor (IMU, Equisense motion S © Equisense, Bidart, France) [29,30] that was placed around the girth, exactly in the middle between the front legs of the horses. Blood samples were collected by venipuncture in the left or right jugular vein using a sterile 2 mL syringe and 23-gauge needle to determine plasma lactate concentrations (LA, mmol/L; Lactate Pro 2^®^ Arkray Inc., Kyoto, Japan).

### 2.3. Data Collection

#### 2.3.1. Preparation Period

During the preparation period, the trainer of each stallion filled out a weekly questionnaire to obtain their training protocol. The data collected were the type of training (lunging, dressage, hacking, driving, other), frequency of training sessions, and duration of a training session.

#### 2.3.2. 70-Day Test

Before all training activities, horses were equipped with an IMU sensor to measure the duration of the training and time spent in each gait. In addition, the type of training was registered (riding, lunging, driving). If the IMU (duration) data from a training was lost, measurements were estimated using the average duration from the training sessions of the same type of that specific horse in the same period. In week 1, week 6, and week 10 of the 70-day test, a SET was performed.

#### 2.3.3. Standardized Exercise Tests

The SETs consisted of three incremental steps and was identical to the SET used in part one [27]. During the SETs, the duration of the exercise, time spent per gait, HR, and LA were measured. Blood samples to determine the LA concentration were taken between 60 and 90 s after finishing each step of the SET, as well as after 10 min of recovery at walk. From the data obtained during the SET, calculations were performed to determine the relationships of the LA to HR as an exponential regression curve. The HR_LA2_ and HR_LA4_ (HR at LA of 2 and 4 mmol/L) were determined by interpolation. The recovery HR after 10 min was determined after the last exercise step.

### 2.4. Data Analysis

External workload was determined as the duration of each training and time spent at a certain gait in minutes (walk, trot, and canter). Before processing, all HR data were visually checked for artefacts. If many artefacts were present (>5% of the measurement), the data were not used for analysis. For HR analysis in the different gaits, the mean HR of the last 60 s in each gait and after 5 and 10 min of recovery were used. For the total mean HR of an exercise, all HR data were used from the start of exercise until after recovery. Horses with the age of 4 and 5 years were combined into one age group for the statistical analysis. The statistical analysis was performed using R studio^®^ (Boston, MA, USA).

#### 2.4.1. Number of Training Sessions

The number of training sessions (total and dressage) ware analyzed based on the number of training sessions per week, with a maximum of 7 training sessions in 1 week. The number of training sessions were modelled with a logistic regression (grouped binominal data), with random horse intercepts and with the period in the study as a predictor variable.

#### 2.4.2. Workload and SET Data

Linear mixed effect models were used for the external workload (duration) and SET data. Normal probability plots of the residuals were created; if the normality assumption did not hold, a log transformation was performed. For the external workload, the horse ID was a random effect, and the period in the study and age of the horse were fixed effects. For the SET data (HR, LA, HR_LA2_, and HR_LA4_)_,_ The LA results were log-transformed to obtain a normal distribution. The horse ID was a random effect, and gait (mean HR, trot, canter 1, canter 2, and recovery after 10 min), SET (I, II or III), age, rider, and gait–SET interaction were fixed effects. Akaike’s Information Criterion (AIC) was used for model reduction. For important effects in the final model, 95% confidence intervals (95% CI) were calculated and presented as the estimate and 95% CI. Log-transformed data are presented as ratios. All data are presented as mean ± s.d.

## 3. Results

### 3.1. 70-Day Test

In total, 16 horses were accepted for the 70-day test. Of these horse, five stallions completed the 70-day test and were approved as a Friesian studbook stallion. Eleven horses were eliminated during the 70-day test; three were eliminated because of lameness, and eight horses were eliminated by the judges.

### 3.2. External Workload

In the preparation period, data from 418 training sessions ware obtained by the questionnaire. A total of 424 training sessions were acquired by the IMU sensor during the 70-day test. Data from 87 training sessions were lost due to technical issues. In the preparation period, the horses were trained 4.4 ± 0.8 times a week with a mean duration of 107 ± 25 min per week (see Table 2). In the weeks between SET-I and SET-II, the horses were trained 4.3 ± 0.7 times a week. The horses were trained less frequently in the weeks between SET-II and SET-III compared with the preparation period (estimate −0.6; 95% CI −0.9, −0.3), but there was no difference between the preparation period and the period between SET-I and SET-II.

According to the AIC, the age of the horse and period of the study were essential factors that affected the total duration of training per week. The horses were trained longer (total minutes per week) in the period between SET-I and SET-II compared with the preparation period (estimate 7; 95% CI 1, 14). However, the training duration per week was shorter in the period between SET-II and SET-III compared with the preparation period (estimate −17; 95% CI −27, −7) and the period between SET-I and SET-II (estimate −24; 95% CI −34, −14). Overall, older horses were trained longer than 3-year-old horses (estimate 13; 95% CI 2, 25). The time spent at trot was not different between the different periods in the study. The horses did less walking (estimate −17; 95% CI −24, −11) and less cantering (estimate −3 min/week; 95% CI −5, −1) in the period between SET-II and SET-III compared with the period between SET-I and SET-II.

### 3.3. Standardized Exercise Tests (SETs)

During the 70-day test 16, 9, and 6 horses participated in SET-I, II, and III, respectively. The ambient temperature on the days of the SETs were 16 °C on the day of SET-I, 12 °C on the day of SET-II, and 9 °C on the day of SET-III. The HR and LA results from SET-I, II, and III are shown in Figure 2 and Figure 3. According to the AIC, the fixed effects of age of the horse, rider, and gait–SET interaction could be left out of the final model, having no important effect on the HR and LA of the horses. In all SETs, the HR (bpm) in canter 1 (estimate 28; 95% CI 24, 31) and canter 2 (estimate 29; 95% CI 25, 32) was higher compared with the HR in trot. Overall, the HR in SET-III was lower than in SET-I (estimate −6; 95% CI −10, −1).

According to the AIC, the gait and SET significantly affected the LA values of the horses. The Log(LA) (mmol/L) was higher after canter 1 (estimate 1.1, 95% CI 2.7, 3.4) and canter 2 (estimate 1.2, 95% CI 2.9, 3.6) compared with Log(LA) after trotting. The recovery Log(LA) did not differ from the LA after trotting. Overall, the Log(LA) of SET-I was not different from SET-II but was significantly less in SET-III (−0.5, 95% CI 0.6, 0.7). The data did not show differences in HR_LA2_. HR_LA4_ (bpm) was higher in SET-III compared with SET-I (estimate 10; 95% CI 0, 20).

## 4. Discussion

In contrast to the original studbook training program (part one of this study), the adapted training program used in this study (part two) showed a positive fitness effect after 70 days of training in young Friesian stallions. In this study, horses had lower HR and LA results in SET-III compared with SET-I, indicating fitness improvement. In the first part of this study (part one), the original training program caused effects on fitness parameters, which are indicative of non-functional overreaching or overtraining in these young horses. Part two showed that relatively small adaptations to a training program for young horses could reverse the physical response of horses from fitness decrements towards fitness improvement. This study showed the importance of a well-thought-out training schedule to prevent inappropriate training loads in young horses with negative training effects.

In part one of the study [27], a group of young Friesian stallions was monitored for their physiological response in the same manner as in the present study. They were housed in the same facility, trained by the same staff, and performed similar SETs and workload assessments. Furthermore, the selection methods (inclusion criteria) were identical. At the start of the 70-day test at SET-I, the fitness of the horses was comparable. The horses in the part one study (original training program) had an average HR at canter 2 of 148 ± 17 bpm versus 154 ± 10 bpm in the part two study (adapted training program). The LA after canter 2 was 3.2 ± 1.4 mmol/L versus 3.5 ± 1.1 mmol/L in part one and part two, respectively. Therefore, the authors have concluded that the horses showed no difference in fitness levels at the start of the studies.

Training frequency during the preparation period, when the horses were trained by their own trainer, seemed to be comparable in both studies. The horses were trained 4.5 ± 0.9 (Part one study) and 4.4 ± 0.8 (Part two study) times a week. However, when evaluating the total estimated training duration per week during the preparation period, it was shown that horses in the present study were trained for a shorter duration compared with the part one study (107 ± 25 min versus 122 ± 25 min of training per week, respectively). Evaluating the results of SET-I, it can be concluded that the fitness of the horses was not affected by the shorter total training time per week (on average, 3 min per training session shorter) in the preparation period during the present study compared with the part one study. As data from training sessions in the preparation period were obtained using weekly surveys filled out by the horse’s trainers, this might have led to a bias. Quantifying workload using surveys is less accurate than actually monitoring the workload of horses on a daily basis [31]. Unfortunately, it was not possible to measure the actual workload from all horses (*n* = 43) during the preparation period as during that period, it is not yet known which stallions will be selected by the judges to enroll in the 70-day test. In addition, the KFPS studbook promised to share the results of the part one study with the trainers, judges, and other professionals involved after finalizing this study. It was only decided afterwards to proceed with a follow-up study using an adapted training program to compare the effects. As it was shared that the training program for the stallions was leading to decreased fitness in the part one study, it is possible that the trainers tried to avoid this and trained shorter or filled in the survey a bit more positive by reducing the training times. However, trainers were not informed beforehand what adaptations were going to be made to the training program of the stallions, nor were they advised as to how to adapt their own training program during the preparation period.

To identify what caused the inverted physiological effects of horses to the adapted training program, the workload that the horses performed in both studies needs to be compared in detail. The 70-day test consisted of two periods; the period between SET-I and SET-II (week 1–6) and the period between SET-II and SET-III (week 7–10). The external workload regarding the training frequency per week did not differ between the preparation period and the 70-day test in the part one study (between 4.4 and 4.7 times a week, respectively). However, the training frequency in the part two study showed a reduction in the second part (period between SET-II and SET-III) of the 70-day test (4.4 versus 3.8 times per week respectively). In addition, the total training time per week during the 70-day test (period between SET-I and SET-II and period between SET-II and SET-III) was longer in the part one study compared with the part two study (144 ± 21 and 131 ± 39 min per week in part one versus 115 ± 20 and 92 ± 32 min per week in part two, respectively). The training time was mainly decreased due to shorter walking periods per training (94 ± 22 and 82 ± 28 min per week in part one versus 71 ± 16 and 55 ± 23 min per week in part two) and canter periods (10 ± 3 and 8 ± 6 min per week in part one versus 8 ± 3 and 5 ± 6 min per week in part two, respectively). The duration of trotting was similar in both studies. Although the difference in training frequency (less than one session per week) and duration (less than 5 min per training) was relatively small, there was a large beneficial effect on the fitness parameters. In addition, an important factor that could explain the improved training results in part two of the study is that the intensive training sessions were always followed by one or two days of relative rest, allowing for recovery. In part one of the study, several intensive training sessions were performed on consecutive days. Training studies in racehorses also show the importance of recovery days in a well-balanced training program that aims to prevent overtraining [20,21,25,26]. Lindner et al. showed that horses undergoing intense training sessions every other day for 6 weeks developed signs of overtraining, but horses undergoing the same training intensity but with two recovery days in between had improved fitness [20,21]. This emphasizes that there is a narrow balance between an appropriate training program versus the risk of nonfunctional overreaching or overtraining.

A contrary finding between part one and part two is that training duration in 4-year-old (and older) horses was shorter than in 3-year-old horses in part one, but longer in the part two study. However, when comparing the absolute values of training duration between both studies, it was shown that 4-year-old horses were trained for similar durations in both studies: the mean training duration per week for all horses in part one was 144 min, and it was 115 min in part two. Four-year-old horses were trained 16 min per week less in part one and 13 min per week more in part two. Therefore, 4-year-old horses on average trained 128 min per week in both studies. The large reduction in the training duration of the part two study seems to be mainly caused by a reduction in the training duration of the 3-year-old horses. Although 4-year-old horses were trained for the same duration in both studies, this was not reflected in the HR or LA values measured in all SETs. The HR and LA values in the SETs of both studies were not affected by the age of the horses. Although the age effect was not statistically important, this might be caused by the low number of older horses, especially in SET-III, as there were less older horses participating in both studies compared with 3-year-old horses. Evaluating the HR and LA values of older horses compared with younger horses showed that in part one, 4-year-old horses had a lower mean HR after canter 2 than 3-year-old horses (144 bpm vs. 160 bpm). In part two, there was less of a difference between the age groups (HR after canter 2 of 154 bpm in 3-year-old horses versus 146 bpm in older horses). In part one, the LA results after canter 2 were similar in both age groups. However, in part two, the mean LA was lower in 4-year-old horses (2.2 mmol/L) compared with 3-year-old horses (3.3 mmol/L). Although not statistically proven, older horses seem to have lower HR and LA results compared with 3-year-old horses.

In part two of the study, the average training duration was 27 min (period between SET-I and SET-II) and 24 min (period between SET-II and SET-III) per session compared with 31 and 27 min in part one, where the horses showed signs of overtraining. Advanced dressage horses were trained 48 min per training session in the descriptive study of Veldman and Rogers [32]. In the descriptive study of Lönnel et al., advanced jumping horses were trained for 19–52 min per training session [33]. Training level and experience are not comparable between the advanced dressage and jumping horses and the young Friesian stallions. In the training study of standardbreds of de Graaf Roelfsema et al., training sessions were 28–37 min [25] and led to improved fitness when horses had four training sessions per week. When comparing these results, it seems that the young Friesian stallions should be trained for a shorter duration than young standardbreds or experienced sport horses.

In the study of de Graaf-Roelfsema et al., standardbreds were trained at ~80% (~192 bpm) of their estimated maximal HR (HR_estmax_, 240 bpm) for three or four bouts of 2–3 min. The horses showing signs of OTS were trained six or seven times a week for 6 weeks in contrast to the horses in the control group, which were trained four times a week. The maximum LA values in the SETs (performing 20 min of exercise at ~80% of HR_estmax_) after the intense training period were 4.9 and 3.2 mmol/L for the control group and the overtrained horses, respectively. The horses with OTS had lower LA results than the control group because the horses had to trot until fatigue in the SET, and the overtrained horses had a much shorter time until fatigue than the horses in the control group [25]. The Friesian stallions in both part one and part two had a lower peak HR (~145–175 bpm) in the SETs than the standardbreds but had comparable LA results (~3–6 mmol/L) when performing canter in an indoor arena. The normal canter in the young Friesian stallions led to similar LA values as standardbreds trotting at 80–85% of their maximal HR. In addition, The HR_LA4_ in the Friesian stallions in both part one and part two (161–170 bpm) was lower than in the trained event horses (189 ± 10 bpm) [34]. Friesian horses may thus reach the anaerobic threshold at a lower HR than standardbreds and competing event horses. Cantering may be seen as moderate- to high-intensity training in young Friesian horses, and the internal load of canter in a young Friesian horse is comparable to standardbred trotters running at 80–85% of their maximal HR. The HR_LA4_ was higher in SET-III compared with SET-I in the present study; thus, the horses reached the anaerobic threshold of 4 mmol/L at a higher HR in SET-III than in SET-I. This might be because of improved fitness in the horses, allowing for higher internal loads before the LA starts accumulating. 

Behavioral changes are an important factor in the overtraining syndrome. Objective behavioral scoring was not performed in part one or part two, and no large mood changes or unwillingness was seen in the horses of part one. However, the same riders were riding the horses in both parts of the study, and they had the impression that the stallions in part two were more energetic and willing to go forward during the 70-day test compared with part one, where the horses felt “tired” in the middle of the 70-day test.

In both studies, the external workload was higher during the 70-day test compared with the preparation period. In part one of the study, the external workload increased by 18% from the preparation period to the first period of the 70-day test. In the present study, the increase was 7% in the same period. In human studies, a high rate of change in workload is associated with increased injury risk [35,36,37,38]. In the present study, a 7% increase in the external workload, combined with alternating intensive training sessions with recovery days, appears to reduce the risk of overtraining compared with the 18% increase in the external workload in part one of the study. In part two of the study, three horses (19%) were eliminated due to an orthopedic injury compared with five horses (31%) in part one. This reduction in injuries might also be related to the adaptations made in the training program. However, the number of horses included in both studies were too low to make definite conclusions on the injury risk related to the measured training load.

Ambient temperature in the SETs was lower in SET-III compared with SET-I, and this might have contributed to the decreased workload of the third SET and thus the lower HR and LA results. However, in part one of the study, ambient temperatures were similar in SET-II and SET-III, where the horses had higher HR and LA results. The impact of the lower ambient temperatures thus seems to be of minimal importance on the measured fitness parameters.

In the 70-day test, the external workload was measured using an IMU sensor, which gave very precise data on the duration and time spent in different gaits [30]. However, technical failure led to missing data from 87 of the 424 training sessions (21%). The main reasons for the failure of measurement were riders that forgot to use the system, batteries not being charged, or technical failure of the phone applications that were used to start and stop the measurements. A calculated replacement for these missing training sessions was used; however, this of course provides less accurate data. The authors are aware of the partial bias that is created due to the data replacement. Replacing data loss with horse and training specific averages seemed to be the most correct solution to this issue as this is also performed in human studies regarding workload assessment. Quantifying internal workload by measuring HR during all training sessions instead of only during SETs would have given more information on the impact of each training session on the horse; however, technical issues prevented these measurements.

The fitness of the Friesian stallions at the start of both part one and part two of this study was enhanced compared with the studies on young Friesian horses from Munsters et al. and de Bruijn et al. [39,40]. The SETs used in all four studies are comparable. When comparing the studies of Munsters et al. and de Bruijn et al. to the result of the stallions in both part one and part two of the present study, the horses in their studies had a higher HR after cantering in the first SET. In the first SET, the horses in the study of de Bruijn et al. also had higher LA results compared with the present study. The LA was not measured in the first SET in the study of Munsters et al. The Friesian stallions in the present study might have been better prepared for a SET. The young stallions have been through several selection tests in the months prior to the 70-day test, for which they already need a certain amount of schooling and training. Potential studbook stallions are often trained by very experienced and professional owners that take the training of these horses very seriously. In the study of Munsters et al. young mares were used, which were included in an offspring performance test. These mares are often not broken in yet before the start of the test, or only lightly trained, and this may explain the lower fitness level at the start of that study.

The present study shows that it is important to consider the daily training workload of young Friesian horses. There is a narrow balance between effective training (part two) and too intense of a training program (part one), and for a subgroup of the young Friesian horses, riding in canter is already a moderate to intense work-out. Using an appropriate training program that allows fitness to improve and prevents non-functional overreaching or overtraining is important for the welfare of sport horses. Using HR sensors and SETs are helpful for building up a safe and effective training program, as horses do not show the internal load of a given exercise. As a translation to the daily practice for trainers of young Friesian horses, days of more intense training should be alternated with resting days that can, for example, consist of low intensity lunging, paddock turn-out, or riding without cantering to allow for tissue recovery and supercompensation to occur. Young Friesian horses are well capable of performing basic dressage and driving training, as long as the training intensity is carefully monitored.

## 5. Conclusions

Training young Friesian stallions three to four times a week with an average duration of 24 min per training session, with alternating intensities per training session, for 10 weeks improved the fitness parameters of HR and LA after cantering during standardized exercise tests. It is important while training young Friesian stallions to alternate the training days with days of active rest or low intensity training (no canter) to reduce the risk of overreaching or overtraining. A well-balanced training program improves the welfare and performance of the equine athlete. Young Friesian horses are well capable of performing dressage and driving training, as long as the training intensity, frequency, and duration is monitored and adjusted meticulously.

## Figures and Tables

**Figure 1 animals-13-00658-f001:**
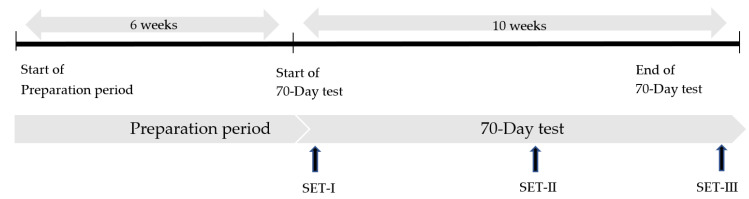
Training schedule of young Friesian horses during the preparation period and the 70-day test, including standardized exercise tests (SETs).

**Figure 2 animals-13-00658-f002:**
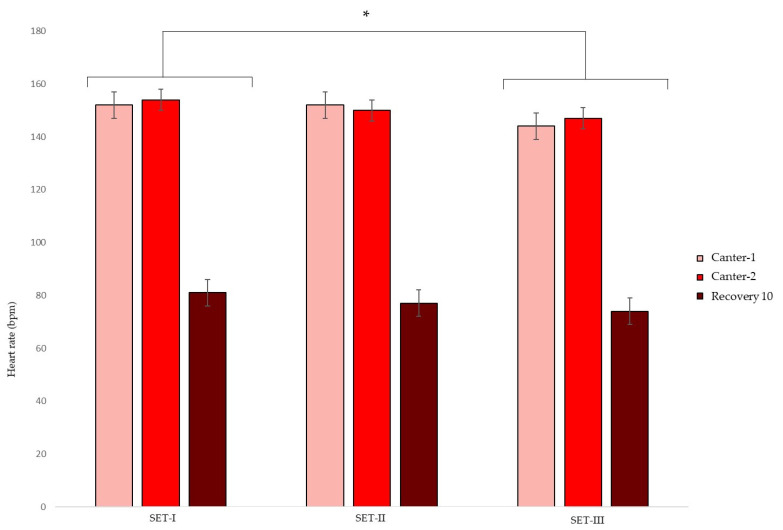
Mean ± s.d. heart rate (bpm) during three consecutive standardized exercise tests (SET-I, SET-II, and SET-III) in young Friesian stallions with 5 (SET-I -SET-II) and 4 (SET-II–SET-III) weeks in between. * Indicating an overall statistical difference from SET-I.

**Figure 3 animals-13-00658-f003:**
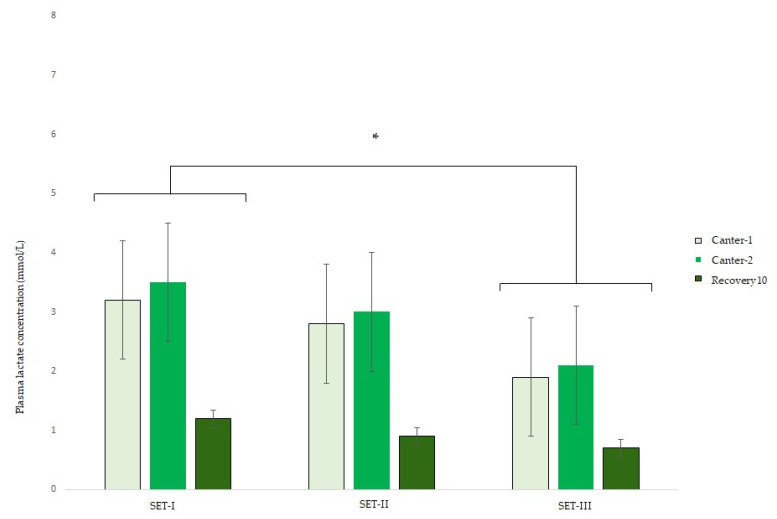
Mean ± s.d. plasma lactate concentration (LA) during three consecutive standardized exercise tests (SET-I, SET-II and SET-III) in young Friesian stallions with 5 (SET-I -SET-II) and 4 (SET-II–SET-III) weeks in between. * Indicating an overall statistical difference from SET-I.

**Table 1 animals-13-00658-t001:** Example of the weekly training schedule of the young Friesian horses during the 70-day test.

Day of the Week		
	Period between SET-I and SET-II	Period between SET-II and SET-III
Monday	Dressage training	Driving training
Tuesday	Low intensity training	Paddock turnout
Wednesday	Dressage training	Driving training
Thursday	Low intensity training or no training	Paddock turnout
Friday	Dressage training	Driving training
Saturday	Paddock turnout	Paddock turnout
Sunday	Paddock turnout	Paddock turnout

**Table 2 animals-13-00658-t002:** External workload (duration and number of training sessions) per type, gait, and per week in young Friesian stallions during the preparation period (*n* = 6 weeks) and 70-day studbook approval test); n.a. = not available.

	Preparation Period		70-Day Test			
			Period between SET-I and II		Period between SET-II and III	
	Sessions/Week	Total Number	Sessions/Week	Total Number	Sessions/Week	Total Number
Total	4.4 ± 0.8	418	4.3 ± 0.7	313	3.8 ± 0.8	111
Dressage	2.6 ± 0.9	250	2.9 ± 0.5	210	1.8 ± 2.0	51
Lunging	1.1 ± 0.7	103	0.7 ± 0.5	48	0	0
Driving	0.7 ± 0.7	65	0.8 ± 0.5	56	2.0 ± 1.4	60
Duration (min/week)						
Total	107 ± 25		115 ± 20		92 ± 32	
Walk	n.a.		71 ± 16		55 ± 23	
Trot	n.a		36 ± 9		33 ± 6	
Canter	n.a.		8 ± 3		5 ± 6	

## Data Availability

Not applicable.

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
