# Peer review of "Longitudinal Training and Workload Assessment in Young Friesian Stallions in Relation to Fitness, Part 2—An Adapted Training Program"

_animals, 2023, doi:10.3390/ani13040658_

Round 1

Reviewer 1 Report

Dear authors, the topic of your work “Longitudinal Training and Workload Assessment in Young 2 

Friesian Stallions in Relation to Fitness: Part 2—an Adapted 3 Training Programme 

” it is very scientific importance in equine exercise physiology and functional selection of Friesian breed, and importance for selection. 

I would like the following considerations to the authors:

 This study is very good but the second part is redundant as it is very similar to the first part. It does not include any previous or final analysis of any animal, nor variables other than part 1. I believe that the two parts should be brought together in a single article, which would have much more strength and that the scientific quality of this journal Animals (Q1) should provide all the information together.

Author Response

Dear reviewer, thank you for your comments and feedback. We tried to assess all points as good as possible. Please find our answers below marked in red.

Dear authors, the topic of your work “Longitudinal Training and Workload Assessment in Young 2 

Friesian Stallions in Relation to Fitness: Part 2—an Adapted 3 Training Programme 

” it is very scientific importance in equine exercise physiology and functional selection of Friesian breed, and importance for selection. 

I would like the following considerations to the authors:

This study is very good but the second part is redundant as it is very similar to the first part. It does not include any previous or final analysis of any animal, nor variables other than part 1. I believe that the two parts should be brought together in a single article, which would have much more strength and that the scientific quality of this journal Animals (Q1) should provide all the information together.

Dear reviewer, thank you for your comments. We understand your suggestion to combine part one and part 2  to one study, and we considered carefully whether both parts should be presented combined or separated. However, we find the two parts are distinct studies. Although the materials and methods are very similar, two different groups of young stallions were used in both parts in two different research set-ups. In part 2 the training programme was adjusted based on findings in part 1. Part 2 was truly a sequel to part 1(with other horses) because of the negative effects on heart rate and plasma lactate concentrations during the SETs in part 1. After these findings the studbook asked the authors for advice how to optimize the training protocol for the young stallions and then re-evaluate the effects of these adaptations and that was done in Part-2. 

Reviewer 2 Report

With interest, I have been reading the manuscript entitled: "Longitudinal Training and Workload Assessment in Young Friesian Stallions in Relation to Fitness: Part 2 " by Siegers et al.

I have revised the Part 1 of this manuscript, and, as stated before, I do not conviced that the 10-week training programme (70-10 Day test) induced overtraining (OT). In this part 2, the researchers have proposed a reduction in external training loading, especially in the training volume, fewer training sessions and fewer minutes of canter per week (L-175, “frequency and duration of training sessions were reduced”). Thus, I ask you. How were rest and recovery phases tempered during the training schedule since that this approach may have tremendous benefits for horses?

Suggestions for improving the manuscript:

Abstract:

L12: “training programme induced overtraining in young horses”.

My comments: Again, I do not convinced that these horses had overtraining. Please, look my comments below.

Introduction

 L 64-66: “During standardized exercise tests horses perform less than before, heart rates (HR) and plasma lactate concentrations (LA) after the same amount of exercise may be increased [4,6]”

My comments: These references are outdated. De Graaf-Roelfsema et al. (2007) mentioned the classical concept of plasma lactate and OT. Low plasma lactate during submaximal SET may indicate OT, the so-called “lactate paradox”. So, could we speculate that these horses presented some carbohydrates/glycogen deficiency or depots are not full, and the formation of lactate is disturbed, and different lactate curves may result (Part-2)? Again, you do not have enough information to say that these horses were in overtraining status (Part-1).

As I stated, we could classify the horses belonging to Part 2 as “lactate paradox horses”. Maybe not.

In Part 1, although the horses showed a rise in blood lactate and heart rate, they completed SET III. Thus, could they be presenting the functional overreaching (FOR)? The time course of the decrement in performance and subsequent supercompensation has been used to discriminate between the distinct phases of the fitness-adaptive fatigue continuum titled functional overreaching (FOR), non-functional overreaching (NFOR) or overtraining syndrome. The short-term transient training-induced decrements in performance evoked by boosted training load (i.e. FOR) are thought to be an acceptable and inevitable element of a training programme. They are often consciously induced in training to stimulate relevant physiological adjustments and performance supercompensation.          

 Discussion

Lines 315:the original training programme caused overtraining in these young horses”

My comments: As aforementioned, authors must bring the functional overreaching (FOR) approach to the manuscript and discuss/expand it in both parts (Part 1 and Part 2). 

Author Response

Dear reviewer, thank you for your comments and feedback. We tried to assess all points as good as possible. Please find our answers below marked in red.

With interest, I have been reading the manuscript entitled: "Longitudinal Training and Workload Assessment in Young Friesian Stallions in Relation to Fitness: Part 2 " by Siegers et al.

I have revised the Part 1 of this manuscript, and, as stated before, I do not conviced that the 10-week training programme (70-10 Day test) induced overtraining (OT). In this part 2, the researchers have proposed a reduction in external training loading, especially in the training volume, fewer training sessions and fewer minutes of canter per week (L-175, “frequency and duration of training sessions were reduced”). Thus, I ask you. How were rest and recovery phases tempered during the training schedule since that this approach may have tremendous benefits for horses?

Dear reviewer, thank for your valuable comments and pointing out your concerns about the presence of overtraining in the horses in Part-1 of the study. In part-1 of the study horses performed worse each following SET demonstrated by increases in HR and LA when performing the same exercise duration and speed in each SET. This decrement in performance lasted for more than 4 weeks, which indicates overtraining according to the paper of Meeusen et al 2013 and Carrard et al 2022, Rivero 2007 and Rivero 2008. We have not objectively measured behavioural changes or body weight in part-1 of the study, however from anecdotal evidence riders indicated that the horses participating in Part-2 study felt less fatigued during training sessions. The horses were more energetic and forward, especially in the middle part of the 10-week training programme. The same riders were riding the horses in Part-1 and Part-2. Thus, in addition to the increase in effort when performing the same type of exercise there seems to be a status of nonfunctional overreaching or overtraining in the horses of Part-1. Throughout the manuscript we now state the term overtraining less direct/strict and used terms as “functional overreaching and nonfunctional overreaching”

In part-2 of the study we made changes in the training programme to reduce training load. Riders and the responsible trainer of the studbook were instructed precisely how often they should train the horses and which training sessions should include canter (moderate to high intensity training), and which training sessions should only consist of walk and trot (low intensity). These instructions were based on the principle of training: giving an training stimulus followed by 1-2 days of relative rest to recover. Studies in other breeds demonstrated improved fitness when 1 or 2 days of recovery were given after a more intense exercise Therefor we agree with the reviewer and other studies that the way how rest and recovery periods were distributed throughout the week is an essential factor leading to effective training or overreaching/overtraining. In addition, using the IMU sensor it was measured what exercise was performed by the horses and thus it was possible to see whether our instructions were followed by trainer and riders. The training protocol is presented in table 1 and in the discussion the information about training and rest period is included

Suggestions for improving the manuscript:

Abstract:

L12: “training programme induced overtraining in young horses”.

My comments: Again, I do not convinced that these horses had overtraining. Please, look my comments below.

Thank you for your comment, the authors agree that horses could be either in nonfunctional overreaching or could be overtrained. We have changed the sentence to “Part-1 of this study showed that the original training programme was too intense and lead to reduced fitness.” In the simple summary,  to  “Part-1 of this study showed that the 70-Day test was too intense and lead to reduced fitness” in the abstract.

Introduction

 L 64-66: “During standardized exercise tests horses perform less than before, heart rates (HR) and plasma lactate concentrations (LA) after the same amount of exercise may be increased [4,6]”

My comments: These references are outdated.

We thank the reviewer for this comment, we have now added two more recent references about overtraining in the introduction. (Carrard 2022, Meeusen 2012)

De Graaf-Roelfsema et al. (2007) mentioned the classical concept of plasma lactate and OT. Low plasma lactate during submaximal SET may indicate OT, the so-called “lactate paradox”. So, could we speculate that these horses presented some carbohydrates/glycogen deficiency or depots are not full, and the formation of lactate is disturbed, and different lactate curves may result (Part-2)? Again, you do not have enough information to say that these horses were in overtraining status (Part-1).

We thank the reviewer for pointing this out. Plasma lactate can either increase or decrease in overtrained individuals (De Graaf Roelfsema 2007, Rivero 2007, Rivero 2008, McGowan 2002, Meeusen 2012,Carrard 2022). When performing exercise tests where (race) horses have to run to fatigue (usually on a treadmill) overtrained horses will stop running earlier, have lower speeds and have lower LA results compared to fit horses. However in submaximal exercise tests that are used in the field in riding/warmblood horses plasma LA after the same exercise is higher in horses that are less fit than in fit horses. When results at SETs decrease after a training period, this is most likely because training was not sufficient, horses have underlying disease or horses are overtrained. 

As I stated, we could classify the horses belonging to Part 2 as “lactate paradox horses”. Maybe not.

In Part 1, although the horses showed a rise in blood lactate and heart rate, they completed SET III. Thus, could they be presenting the functional overreaching (FOR)? The time course of the decrement in performance and subsequent supercompensation has been used to discriminate between the distinct phases of the fitness-adaptive fatigue continuum titled functional overreaching (FOR), non-functional overreaching (NFOR) or overtraining syndrome. The short-term transient training-induced decrements in performance evoked by boosted training load (i.e. FOR) are thought to be an acceptable and inevitable element of a training programme. They are often consciously induced in training to stimulate relevant physiological adjustments and performance supercompensation. 

We thank the reviewer for these comments. If we follow the most recent guidelines for overreaching/overtraining in human athletes, in functional overreaching the decrease in performance lasts for approximately 2 weeks. We agree that this can be an acceptable and functional element of a training programme. The stallions in Part-1 showed a decreased performance for a longer period than 2 weeks, since in the second SET (week 6) and third SET (week 10) horses performed worse compared to the first SET (week 1). The authors think they have enough evidence to classify the horses as nonfunctional overreaching or overtraining, we do agree that we have cannot make the distinction between NFOR or OT. Thus we have changed this throughout the manuscript.

 Discussion

Lines 315: “the original training programme caused overtraining in these young horses”

My comments: As aforementioned, authors must bring the functional overreaching (FOR) approach to the manuscript and discuss/expand it in both parts (Part 1 and Part 2). 

We thank the reviewer for this comment. The sentence is now changed to: “the original training programme caused nonfunctional overreaching or overtraining in these young horses”. We have added FOR and NFOR throughout the manuscript.

Reviewer 3 Report

The manuscript "Longitudinal training and workload assessment in young Friesianstallions in relation to fitness: Part 2 – an adapted training programme" focus on a very interesting and actual subject regarding a correct training programme of young horses and welfare of sport horses.

The Abstract is correct and elucidates the content of the manuscript.

The introduction section seems a little bit too long. Lines 56-57 “Is the next intense exercise repetitively performed during the recovery phase the risk of overtraining increases”. I do not understand this sentence.

The materials and methods seem adequate. Line 198. The horses age of 3 (n=11), 4 (n=3), or 5 (n=2) years in the study is a concern, but the authors try to justify and explain…

The results are sound. In Table 2, the number of training sessions should be written in the left column (Total, Dressage, Lunging, Driving number of training sessions).

The discussion is correct. Line 368 (less than 1 session per week).

Author Response

Dear reviewer, thank you for your comments and feedback. We tried to assess all points as good as possible. Please find our answers below marked in red.

The manuscript "Longitudinal training and workload assessment in young Friesian
stallions in relation to fitness: Part 2 – an adapted training programme" focus on a very interesting and actual subject regarding a correct training programme of young horses and welfare of sport horses.

The Abstract is correct and elucidates the content of the manuscript.

The introduction section seems a little bit too long. Lines 56-57 “Is the next intense exercise repetitively performed during the recovery phase the risk of overtraining increases”. I do not understand this sentence.

We like to thank the reviewer for this comment. We have shortened the introduction by shortening some parts of the introduction. The mentioned sentence is now changed to “Performing intense exercise repeatedly before the recovery phase is finished increases  the risk of overtraining”.

The materials and methods seem adequate. Line 198. The horses age of 3 (n=11), 4 (n=3), or 5 (n=2) years in the study is a concern, but the authors try to justify and explain…

We thank the reviewer for pointing out this concern and we agree that the age distribution of the horses was not ideal. However in this study in a real life situation, using privately owned horses selected for participation by the studbook judges, we could not influence which horses would participate. We did comment on this in the discussion, age was also included in the statistical model to correct for this issue.

The results are sound. In Table 2, the number of training sessions should be written in the left column (Total, Dressage, Lunging, Driving number of training sessions).

We thank the reviewer for this comment and have added the numbers of the training sessions in the table

The discussion is correct. Line 368 (less than 1 session per week).

We thank the reviewer for this comment and the sentence is adjusted

Round 2

Reviewer 2 Report

The authors have addressed all recommendations for revision and therefore the reviewer recommends to accept manuscript for publication.